# Revealing Further Insights on Chilling Injury of Postharvest Bananas by Untargeted Lipidomics

**DOI:** 10.3390/foods9070894

**Published:** 2020-07-08

**Authors:** Juan Liu, Qingxin Li, Junjia Chen, Yueming Jiang

**Affiliations:** 1Guangdong Engineering Lab of High Value Utilization of Biomass, Guangdong Provincial Bioengineering Institute (Guangzhou Sugarcane Industry Research Institute), Guangdong Academy of Sciences, Guangzhou 510316, China; ljane0505@126.com (J.L.); qingxin_li@outlook.com (Q.L.); 2Key Laboratory of Plant Resources Conservation and Sustainable Utilization, Guangdong Provincial Key Laboratory of Applied Botany, South China Botanical Garden, Chinese Academy of Sciences, Guangzhou 510650, China

**Keywords:** banana, chilling injury, untargeted lipidomics, lipid, fatty acids, PLD, LOX

## Abstract

Chilling injury is especially prominent in postharvest bananas stored at low temperature below 13 °C. To elucidate better the relationship between cell membrane lipids and chilling injury, an untargeted lipidomics approach using ultra-performance liquid chromatography–mass spectrometry was conducted. Banana fruit were stored at 6 °C for 0 (control) and 4 days and then sampled for lipid analysis. After 4 days of storage, banana peel exhibited a marked chilling injury symptom. Furthermore, 45 lipid compounds, including glycerophospholipids, saccharolipids, and glycerolipids, were identified with significant changes in peel tissues of bananas stored for 4 days compared with the control fruit. In addition, higher ratio of digalactosyldiacylglycerol (DGDG) to monogalactosyldiacylglycerol (MGDG) and higher levels of phosphatidic acid (PA) and saturated fatty acids but lower levels of phosphatidylcholine (PC), phosphatidylethanolamine (PE), and unsaturated fatty acids were observed in banana fruit with chilling injury in contrast to the control fruit. Meanwhile, higher activities of phospholipase D (PLD) and lipoxygenase (LOX) were associated with significantly upregulated gene expressions of *MaPLD1* and *MaLOX2* and higher malondialdehyde (MDA) content in chilling injury-related bananas. In conclusion, our study indicated that membrane lipid degradation resulted from reduced PC and PE, but accumulated PA, while membrane lipid peroxidation resulted from the elevated saturation of fatty acids, resulting in membrane damage which subsequently accelerated the chilling injury occurrence of banana fruit during storage at low temperature.

## 1. Introduction

Fresh crops after harvest are very perishable. Low temperature storage is frequently used to prolong the postharvest life of perishable fruits and vegetables. However, improper temperature of cold storage could promote quality deterioration and senescence processes. This issue is especially prominent in tropical and subtropical fruits, such as bananas, which are easily subjected to chilling injury when the storage temperature is below 13 °C. This problem could result in short shelf life and great wastage of postharvest banana fruit [1]. Therefore, to elucidate the mechanisms of chilling injury is paramount in exploring a cost-effective solution to extend the shelf life of tropical and subtropical fruits under cold storage condition.

Our previous investigation reported that chilling injury of banana fruit was closely related to energy deficit, oxidative damage and ion concentration changes [2,3,4]. Specifically, the main factors affecting the quality of banana fruit under cold storage are attributed to respiration rate, ATP level, reactive oxygen species (ROS) level, antioxidant content, antioxidant enzyme activities, and H^+^ and Ca^2+^ fluxes. Moreover, elevated electrolyte leakage was observed in banana fruit as chilling injury gradually appeared, which indicated that the membrane integrity was damaged during cold storage. In addition to signaling perception and barrier function, the cell membrane also provides prompt response to internal physiological metabolism under biotic and abiotic stress conditions. Under low temperature condition, the cell membrane usually adapts the composition of membrane itself to maintain normal function. For example, the unsaturation fatty acids could be increased to maintain the membrane fluidity [5]. It has been hypothesized that the phase transition of membrane lipids could be associated with the initial of the chilling injury, which could be estimated by the changes in the compositions of lipids and fatty acids and the unsaturation of fatty acids [6,7]. Moreover, the accumulation of ROS resulting from metabolic disorder caused by chilling injury could induce cellular oxidative stress, and then, membrane lipid peroxidation is consequently triggered ceaselessly, leading to irreversible injury to cell membrane [8].

Lipids are major components of plasma membrane (PM) and act as the interface between cell and environment. The PM has a thick bilayer with tight lipid packing and negative cytoplasmic surface charges for its barrier function [5]. Based on the chemical backbone in combination with distinct hydrophobic and hydrophilic elements, lipids can be divided into main eight groups: fatty acyls, glycerolipids, glycerophospholipids, sphingolipids, sterol lipids, prenol lipids, saccharolipids, and polyketides [9]. Plants sense stimuli and transmit signal transduction into downstream biological responses through the plasma membrane, which is generally the source for signaling lipids [10]. The list of signaling lipids includes phosphatidic acids (PAs), phosphoinositides (PIs), sphingolipids, lysophospholipids, oxylipins, N-acylethanolamines, and free fatty acids [11]. A notable increase in PA, lysophosphatidylcholine (LPC), and lysophosphatidylethanolamine (LPE) is a response to low temperature [12]. Recently, it has been shown that membrane lipid compositions alter significantly during chilling injury of bell peppers at 4 °C while PC, PE, PA, and MGDG are determined to be the crucial biomarkers in the chilling injury process [13]. PC, PE and PG function as the main components of the membrane lipids in relation to the chilling injury of peach fruit during cold storage [14,15]. PA was reported to be the degradation product of the structural phospholipids while the accumulation of PA could further promote the production of free radicals and hydroperoxides and then lead to severe membrane damage [16,17]. Other studies showed that the change in fatty acids of membrane lipids and the declined unsaturation of fatty acids were closely associated with the chilling injury process [18,19]. In addition, a series of investigations indicated that phospholipase D (PLD) played an important role in regulation of membrane lipid composition in response to cold stress [20]. Membrane damage could be initiated by a lipolytic cascade, with PLD and LOX participating in catalyzing phospholipid catabolism [21]. It has been well demonstrated that chilling temperatures promoted the production of ROS, leading to membrane damage in chilling-sensitive fruit [22], whereas MDA is one important parameter for indicating the magnitude of membrane damage caused by ROS [23].

In recent years, the lipidomic methods based on liquid chromatography–mass spectrometry (LC-MS) provided more comprehensive insights about the responses of membrane lipid compositions and functions to chilling [24]. Lipidomic analysis is a powerful method to systematically investigate the biological lipid compositions and constitute changes based on high-throughput analysis. Kong et al. [13] reported that a cell membrane lipidomic approach was able to reveal the participation of membrane lipid metabolism in chilling injury process of bell peppers at 4 °C. Combined lipidomic and transcriptomic analyses have been reported to explore the wax formation with respect to the involvement of membrane lipids in citrus fruit [25]. Chen et al. [14] found that methyl jasmonate promoted phospholipid remodeling to alleviate chilling injury in peach fruit by using untargeted lipidomic analysis. Thus, through application of lipidomic analysis, we can obtain information on lipid families and lipid molecules in various biological processes and can help to elucidate the relevant biological processes. In our study, ultra-performance liquid chromatography–mass spectrometry based lipidomics analysis was applied for lipid profiling while orthogonal projection to latent structure discriminant analysis (OPLS-DA) was performed in order to determine the lipids with significant differences between the banana fruit stored for 0 and 4 days at 6 °C. Moreover, the compositions and relative contents of fatty acids were analyzed. In addition, PLD and LOX activities and their corresponding gene expressions, along with MDA content, were determined. These results provided new insights into the occurrence of chilling injury in relation to membrane lipid metabolism in banana fruit.

## 2. Materials and Methods

### 2.1. Plant Materials and Treatments

Green mature banana fruit were picked from an orchard in Guangzhou, China, and then washed and selected for uniformity of shape, color and size. Fruit was air-dried, packed in plastic polyethylene bags (200 × 150 mm, 0.03 mm in thickness), with 3 fruits per bag and stored at 6 °C and 85–90% relative humidity (RH). Ten fruits were randomly selected after 0 and 4 days of storage. The skin tissues were sliced, frozen in liquid N_2_, grinded into powder by a grinder for 20 s and stored at −80 °C prior to use.

### 2.2. Lipid Extraction and Analysis

Fresh samples (80 mg) were weighed, crushed, extracted by 240 µL precooling methanol solution by vortexing for 2 min, and added with 800 µL MTBE buffer and 200 µL ultra-pure water by vortexing for 2 min. The solution was extracted by ultrasonicating in ice-cold water for 20 min and placed at room temperature for 30 min. The extract was centrifuged at 8000× *g* for 15 min at 10 °C, and the supernatant was collected and dried with nitrogen stream. The dried extraction was dissolved in 200 µL isopropanol by vortexing for 2 min and followed by centrifugation at 8000× *g* for 15 min at 10 °C. The supernatant was subjected to UPLC-QE-MS analysis.

Extracts were analyzed using a Waters ACQUITY UPLC System (Waters, Milford, MA, USA) controlled by LipidSearch™ 4.1 and ACQUITY PDA detector (UPLC eLambda) fitted with ACQUITY UPLC CSH C18 column (2.1 × 100 mm i.d., 1.7 µm, Waters Corp., Milford, MA, USA) and kept at a temperature of 40 °C. The flow rate was 0.3 mL min^−1^. The mobile phase consisted of 10 mM ammonium formate in acetonitrile and water (acetonitrile:water = 6:4, *v/v*) (A) and 10 mM ammonium formate in acetonitrile and isopropanol (acetonitrile: isopropanol = 1:9, *v/v*) (B). The linear gradient elution program was performed as follows: 0–7 min, 30% B; 7–25 min, 30–100% B; 25–30 min, 30% B. The injection volume was 4 µL. Mass spectrometry was recorded using a Q-Exactive Plus (Thermo Scientific™, Waltham, MA, USA) equipped with an ESI source and controlled by Thermo Scientific™ 4.1 software. A full MS scan was performed in the range of *m*/*z* 200–1800 Da at a resolution mode. The resolution was 70,000. The capillary voltages were set at 2.5 kV, and the cone voltage was 40 V. The heater temperature was 300 °C, and the capillary temperature was 350 °C. The spray voltage was 3.0 kV for the positive mode and 2.5 kV for the negative mode. The sheath gas flow rate was 45 arb, the aux gas flow rate was 15 arb, and the sweep gas flow rate was 1 arb. S-Lens RF Level was 50% for the positive mode and 60% for the negative mode.

### 2.3. Lipidomics Analysis

The lipids were recognized and identified by LipidSearch software Version 4.1 (Thermo Scientific™, Waltham, MA, USA). The main parameters were set as follows: precursor tolerance of 5 ppm, product tolerance of 5 ppm, and product ion threshold of 5%. OPLS-DA was applied to the data by using SIMCA-P Software 14.1 (Umetrics, Umeå, Sweden) to discover the lipids differences. The lipids with significant differences were screened between the peel of banana stored for 4 days (chilling injury group) and those stored for 0 d (control group) at 6 °C. The screening criteria were Variable importance for the projection (VIP) > 1 and *p* value < 0.05.

### 2.4. Ingredients and Relative Contents of Fatty Acids of Membrane Lipids

Powdered sample (1 g) was mixed with 5 mL petroleum ether, followed by ultrasonic extraction at 50 °C for 30 min with three replicates. The supernatant was collected and dried with nitrogen stream. The dried sample was mixed with 6 mL n-hexane and 0.5 mol L^−1^ KOH in methanol (*v*/*v* = 1:1) and heated at 60 °C for 60 min in the oven. After cooling down, the mixture was centrifuged at 5000 g for 5 min, and the supernatant was filtered through 0.22 µm membrane. One µL was analyzed using the gas chromatograph (6890, Agilent Technologies Inc., Santa Clara, CA, USA) controlled by a flame ionization detector and equipped with a column (30 m × 0.25 mm × 0.25 µm, Agilent Technologies Inc., Santa Clara, CA, USA). The temperature of the detector was 270 °C. The carrier gas was N_2_ with a velocity of 1 mL min^−1^_._ The flow rates of H_2_ and air were 40 mL min^−1^ and 400 mL min^−1^, respectively. The split ratio was 1:100. The temperature programming of GC oven started from 90 °C, increased to 190 °C with an increasing rate of 7 °C min^−1^, followed by an increase to 215 °C with an increasing rate of 3 °C min^−1^ and maintained for 10 min and increased to 230 °C with an increasing rate of 20 °C min^−1^ and maintained for 5 min. The qualitative analyses of fatty acids were carried out by comparing the retention time to the authentic standard, and the relative amount was expressed as the proportions of total fatty acids.

### 2.5. PLD and LOX Activities

PLD activity was analyzed according to the method described in the reference [26]. Briefly, after PLD extraction by Tris-HCl buffer (pH 7.0), the enzymatic reaction was performed at 28 °C for 1 h with phosphatidylcholine as substrate. Determination of LOX activity was carried out according to the method described by the literature [11]. Briefly, after LOX extraction by PBS solution (pH 7.0) with PVP from peel tissue (5 g), the enzymatic reaction was performed at 30 °C with sodiulinoleate as substrate. PLD activity was analyzed at 520 nm, and LOX activity was measured at 234 nm by using assay kits (Comin Biotechnology Co., Ltd. Suzhou, China) in accordance with the manufacturer’s instructions. One of unit (U) of PLD or LOX activity was defined as a change of 0.001 in absorbance per hour. Results were expressed as U g^−1^ FW (fresh weight).

### 2.6. Determination of Lipid Peroxidation

MDA content was determined according to the literature [27]. Frozen peel tissues (1.0 g) were homogenized in 10 mL of 10% (*w*/*v*) trichloroacetic acid (TCA) and centrifuged at 12,000× *g* for 15 min. The supernatant (1 mL) was added into 3 mL of 0.5% (*w*/*v*) thiobarbituric acid (TBA), and then, the mixture was kept in boiling water for 20 min. The mixture was instantly cooled down in an ice bath and added to 3 mL with 0.5% TBA. The absorbance of the mixture was measured at wavelengths of 450, 532, and 600 nm. The result was expressed as nmol g^−1^ FW.

### 2.7. Total RNA Extraction and cDNA Synthesis

Total RNA was extracted from banana peel using the hot borate method as described by the literature [28]. RNA concentration was determined by measuring UV absorbance at 260 nm (A_260_) with a SmartSpec Plus spectrophotometer (BioRad, Hercules, CA, USA). The integrity and quality of the RNA were verified by agarose gel electrophoresis using the A_260_/A_280_ ratio. The first strand cDNA was synthesized by using the PrimeScriptTM RT Master Mix (TAKARA, Dalian, China) subjected to qRT-PCR amplification, following the manufacturer’s instructions.

### 2.8. Quantitative Real-Time PCR (qRT-PCR)

SYBR Premix Ex TaqTM mix (TaKaRa, Dalian, China) was used as a reaction mixture by adding 10.0 µL of SYBR Premix Ex TaqTM, 0.4 µL of PCR forward primer (10 µM), 0.4 µL of PCR reverse primer (10 µM), 0.4 µL of ROX reference dyeII, and 2 µL (20 ng) of cDNA in a final volume of 20 µL. Relative expression levels were analyzed by the quantitative real-time PCR using the ABI 7500 Real-Time PCR System (Applied Biosystems, Carlsbad, CA, USA) and the Light Cycler 480 SYBR Green I Master Mix (Roche Applied Science, www.roche-applied-science.com) by the manufacturer’s instructions under the following conditions: 30 s at 95 °C, 40 cycles of 5 s at 95 °C, and 34 s at 60 °C. The PLD primers were designed using Primer 6.0 (Primer, Palo Alto, Canada) (forward primer 5′-TGGGAGCCTATCAGCCTCATCATAC-3′ and reverse primer 5′ GCCGTGTGGATGTCTTCTGTTCTT-3′); the LOX primers were designed using Primer 6.0 (forward primer 5′-AACGAGGAAGGCGAAGACCAGTA-3′ and reverse primer 5′-ACCACGATGCTCTCCAGGAAGAA-3′), and the Actin gene was used for quantitative normalization (forward primer 5′-TGGTATGGAAGCCGCTGGTA-3′ and reverse primer 5′-TCTGCTGGAATGTGCTGAGG-3′). The relative expression levels of target genes were calculated using the formula 2^ΔΔCT^. All analyses were repeated three times using three biological replicates.

### 2.9. Statistical Analysis

The results were expressed as means ± standard error (SE). Significant differences were tested by one-way analysis of variance with SPSS 22 (IBM, New York, NY, USA). Statistical differences with *p* value of <0.05 were considered as significant.

## 3. Results

### 3.1. Lipid Analysis by OPLS-DA Model

OPLS-DA is typically used to reveal differences between two groups. VIP is used to explore those variables which exert big impact on the model and is obtained by using SIMCA-P Software 14.1 based on the variable influence on projection. In this study, OPLS-DA model was applied to screen the differences in lipids between chilling injury group and control group. As shown in the score plot (Figure 1A), there were five parallels of each group and two groups were clearly separated. As shown in the loading scatter plot (Figure 1B), PC (34:2), PC (36:4), PC (18:3/18:2), PC (34:3), PC (36:6), and DGDG (16:0/18:2) made significant contributions to the group classification. R^2^Y and Q^2^ are the evaluation parameters of the model. In this model, R^2^Y = 0.998 and Q^2^ = 0.821, indicating that the quality of the OPLS-DA model was excellent to screen the key lipids between two groups.

The results revealed that the OPLS-DA model with satisfactory explanatory and predictive ability was able to screen the crucial biomarkers during the chilling injury process.

### 3.2. Lipid Constitutes with Significant Differences during Chilling Injury

VIP value can reflect the expression mode of all lipids to the extent of influence and explain the ability of classification discriminant of the all grouped samples in the OPLS-DA model. In addition, VIP value is obtained from the multivariate statistical analysis to determine the importance of variables in OPLS-DA model, and it is more comprehensive and important than the *p* value < 0.05, which is obtained from the univariate statistical analysis. We screen those lipids with VIP > 1 and *p* value < 0.05 as key biomarkers with significant differences during chilling injury. When lipids with *p* value < 0.05, those lipids with higher VIP are considered as more important components during chilling injury. Table 1 lists the lipids with significant differences. The PC (18:2/18:3), PC (18:3/18:3), PE (16:0/18:2), and PA (35:2) were the four top lipids with high VIP values, which indicated that these four lipids changed dramatically during chilling injury.

After 4 days of storage at 6 °C, a series of changes occurred in peel of banana fruit. As shown in Table 1, forty-five lipids showed significant changes in peel of banana of control group and chilling injury group. All lipid classes with significant differences were determined. There were three categories including glycerophospholipids, saccharolipids, and glycerolipids. Glycerophospholipids are represented by PE (12 species), PC (8 species), PA (3 species), phosphatidylglycerols (PG) (3 species), phosphatidylinositol (PI) (2 species), phosphatidylserine (PS) (1 species), LPE (1 species), and LPC (1 species). Saccharolipids are represented by MGDG (4 species) and DGDG (3 species). Glycerolipids are represented by triglyceride (TG) (5 species) and diglyceride (DG) (2 species) (Figure 2). The proportions of glycerophospholipids, saccharolipids, and glycerolipids to the number of lipid species that showed significant changes were 69%, 16%, and 15%, respectively. It is worthy to notice that the species of PE and PC accounted for 45% of the significant lipids, and the proportions of LPE and LPC were 2%, respectively.

These results suggested that 45 lipid compounds, including glycerophospholipids, saccharolipids, and glycerolipids, were identified with significant changes in peel tissues of bananas stored for 4 d compared with the control fruit.

### 3.3. Membrane Lipids with Significant Differences during Chilling Injury

Changes in the relative levels of lipids with significant differences from the chilling injury group and control group are shown in Figure 3. Phospholipids, mainly PA, PC, PE, and PG were differentially accumulated in peel of banana fruit with severe chilling injury. The levels of PC, PE, PS, and TG decreased significantly. The levels of PA, PI, PG, LPC, LPE, DGDG, MGDG, and DG increased remarkably. The contents of LPC and LPE as the intermediate products of lipid degradation in chilling injury group were 1.2 and 1.5 times as many as control group. The PA level of chilling injury group was 1.4 times that of the control group. The ratio of DGDG to MGDG increased from 3.5 to 4.2 during chilling injury.

All these data revealed relatively unexpected roles of membrane lipids during chilling injury. Among them, PC, PE, PA, DGDG, and MGDG were significantly changed and consequently were considered as crucial biomarkers during chilling injury.

### 3.4. Changes in Fatty Acids Compositions during Chilling Injury

As shown in Table 2, compared to the control, the relative contents of oleic acid (C18:1), linoleic acid (C18:2) and linolenic acid (C18:3) in bananas under chilling injury decreased significantly, while that of palmitic acid (C16:0) and stearic acid (C18:0) increased significantly during chilling injury. These results indicated that the degradations of unsaturated fatty acids were accompanied by the formations of saturated fatty acids in bananas during cold storage.

### 3.5. Changes in Activities and Gene Expressions of Lipid-Related Enzymes during Chilling Injury

As mentioned above, degradation of membrane lipids, elevated saturated fatty acids, and declined unsaturated fatty acids were found to be involved in chilling injury of banana fruit. Here, to further identify whether membrane lipid metabolism could be induced by cold, the activities and gene expressions of two kinds of key lipid-related enzymes, PLD and LOX were analyzed in this study. As shown in Figure 4, remarkable increases in PLD and LOX activities were observed while *MaPLD1* and *MaLOX2* expressions in banana fruit with chilling injury were significantly upregulated compared to the control fruit.

These findings suggested that lipids degradation in relation to participations of PLD and lipid peroxidation with involvement of LOX promoted continuously the damage of membrane integrity, and therefore, the decline in resistance to cold stress with a series of chilling injury symptoms in banana fruit occurred.

### 3.6. Lipid Peroxidation during Chilling Injury

As shown in Figure 5, the MDA content in banana fruit under chilling injury was 4.7 times that of the control group. This result indicated that the oxidative damage to membrane was severe to promote further physiological disorder and quality deterioration in banana fruit.

## 4. Discussion

### 4.1. Chilling Injury of Postharvest Bananas Was Associated with Degradations of Membrane Lipids

Changes in lipid compositions and structures have been increasingly reported in postharvest of fruits in response to chilling stress [13,14,29]. However, none of these studies displays distinctive differences in postharvest bananas during chilling injury, including lipids degradation, peroxidation, and critical enzymes involved in lipid metabolism, indicating a need for a more comprehensive analysis of lipid changes in postharvest bananas exposed to chilling stress.

In our study, 45 lipid compounds, including glycerophospholipids, saccharolipids, and glycerolipids, were identified with significant changes in peel tissues of bananas under chilling injury in contrast to the control. Among them, the levels of LPC and LPE increased significantly with the extension of storage time. Lysophospholipids as signaling lipids are derived from the hydrolysis of the membrane lipids, then released into the extracellular space and finally can be recognized by the extracellular receptors to initiate signaling pathways [10]. The concentration of lysophospholipids in plant tissue is low, but they will accumulate as the membrane is damaged. In our work, the PA level of chilling injury group was 1.4 times that of the control group. PA could be generated through two pathways: PLD pathway producing PA and substituent (e.g., choline/ethanolamine) via the hydrolysis of glycerophospholipids (e.g., PC and PE) and PLC/diacylglycerol kinase (DGK) pathway [30]. PLD is capable of directly catalyzing the hydrolysis of phospholipids, and then, the hydrolysates can be further hydrolyzed to free fatty acids by lipase [31]. It has been demonstrated that the membrane phospholipid degradation was accompanied with PA accumulation in response to cold stress, leading to membrane damage [16,32]. Our study indicated that there was higher PA level in the chilling injury group compared with the control group. However, the PA level in guava fruit with chilling injury was reduced significantly, and this reduction may be related to the mitochondrial dysfunction involved in the development of chilling injury symptoms [33]. This opposite conclusion may be resulted from the phospholipids extracted from all biological membranes in our work while the phospholipids in guava fruit were obtained from microsomal membranes. Further investigations should be conducted to explore the role of phospholipids in chilling injury symptoms of banana fruit. Our experiment also showed significant decreases in PC and PE levels and a remarkable increase in PA, which suggested that the PLD pathway to generate PA was activated under chilling stress. Similar results were obtained in anthurium cut flowers and bell pepper [13,34]. Recent studies indicated that methyl jasmonate treatment increased the levels of phospholipid biomarkers of peach fruit (especially PC and PE), promoted phospholipid remodeling, and consequently alleviated chilling injury [14].

MGDG and DGDG as structural lipids constitute the skeleton of plastid membrane [35]. It was reported that conical MGDG molecules can be easily converted to cylindrical DGDG to maintain the stability of thylakoid membranes during cold stress [24]. The ratio of DGDG to MGDG increased from 3.5 to 4.2 when subjected to chilling injury in our study. Previous studies have reported that the increase in the ratio of DGDG to MGDG could enhance plant stress tolerance by affecting phospholipids content as well as membrane structure [36]. Recent investigations in green bell pepper treated by methyl jasmonate provided evidence about involvement of lipids in alleviating chilling injury, particularly higher ratio of DGDG to MGDG as well as higher levels of PC and PE [37].

Taken together, our work indicated that membrane lipids were involved more particularly in chilling injury of banana fruit. Furthermore, membrane lipidomic analysis revealed that PA, PC, PE, DGDG, and MGDG were significant biomarkers during chilling injury of banana fruit stored at 6 °C and accompanied of elevated PA level and ratio of DGDG to MGDG. Declined levels of PC and PE were also observed. All these data suggested the participation of membrane lipids in the chilling injury process of banana fruit.

### 4.2. Chilling Injury of Postharvest Bananas Was Associated with Peroxidation of Membrane Lipids

To understand the relationship of lipid peroxidation and chilling injury in postharvest bananas, we measured the fatty acids compositions, activities of key enzymes, and MDA content. In our work, a saturation of fatty acids, enhanced LOX activity, and increased MDA content were observed in peel of bananas with chilling injury, indicating that lipid peroxidation was in close association with chilling injury of postharvest bananas.

Our previous study showed that chilling injury of banana fruit resulted from energy deficit [2]. Energy status was closely associated with the membrane stability, and ATP level was found to have positive association with the ratio of unsaturated to saturated fatty acids in postharvest kiwifruit [38]. It has been hypothesized that the unsaturation of membrane lipids could reflect the chilling tolerance of fruit stored at low temperature [39]. In the recent study, Alba-Jiménez et al. [33] reported that application of 5 kPA CO_2_ atmosphere could improve the unsaturation of membrane lipids and consequently reduce the chilling injury symptoms of guava fruit stored at 10 °C. Yao et al. [40] found that glycine betaine treatment could delay the chilling injury of zucchini fruit stored at 1 °C for 15 days by reducing lipid peroxidation which resulted from the higher levels of unsaturated fatty acids. Recently, Vazquez-Hernandez et al. [41] reported that high CO_2_ treatment alleviated the cell ultrastructure damage by increasing the unsaturation of 18-carbon fatty acids, unsaturation lipid ratio, and index of unsaturated fatty acids and, therefore, reduced lipid oxidative damage in table grapes during long-term cold storage. In addition, higher unsaturation of membrane lipids played an important role in maintaining the citrus fruit quality and in inhibiting litchi pericarp browning [42,43]. Recent investigations in bell peppers provided evidence about involvement of cell membrane lipids in the chilling injury process. Kong et al. [13] found that low temperature stress could trigger membrane lipid metabolisms in bell peppers by application of membrane lipidomic analysis. Ge et al. [19] reported that chilling injury symptoms of bell peppers mostly resulted from cytomembrane destabilization, which was induced by a decrease in desaturation of fatty acids. Ma et al. [37] found that methyl jasmonate treatment could delay the membrane lipid degradation, maintain the membrane structure stability, and therefore delay the chilling injury of green pepper during cold storage. Taken collectively, these reports were in accordance with our results, which supported credence for high unsaturation of membrane lipids to improve chilling tolerance of fruit.

In addition to reveal the changes in lipid compositions, it is also imperative to explore the critical enzymes involved in lipid metabolism. It is well documented that the alteration of membrane lipids could be attributed to the action of membrane lipid-degrading enzymes such as PLD and LOX [44]. PLD activity has been shown to be greatly increased after the loss of membrane integrity [21]; therefore, the decrease in PC and PE levels and increase in PA content could also occur after cell lysis and loss of membrane integrity. LOX can initiate the peroxidation of cell membrane lipids via exclusively catalyzing the oxygenation of polyunsaturated fatty acids generating peroxidation products [45]. Those peroxidation products might attack the cell membrane, resulting in the loss of compartmentalization of cell membrane system [46]. It has been found that the activated PLD and LOX, which participated in lipid degradation and peroxidation, might account for the accelerated developments of chilling injury [47]. The high activities of PLD and LOX were attributed to their gene expression [48] while the increases in activities of these two enzymes were accompanied by the induction of chilling injury [49].

ROS is produced in living organisms by aerobic metabolism during aging process or under stress conditions. However, ROS is highly reactive and initiates lipid peroxidation and causes oxidative damage to chloroplasts, mitochondria, and apoplast under temperature stress [50]. Lipid oxidation has been linked to plasma membrane damage, and thus, excessive ROS might induce structural integrity loss of cellular membrane and cause cellular de-compartmentalization [45]. The polyunsaturated fatty acids that constitute the phospholipids such as PC and PE could be oxidized by ROS, which was generated and accumulated in response to low temperature, and therefore, the membrane fluidity would decline and result in membrane damage. MDA as the end-product of lipid peroxidation in this process is considered as the leading indicator to evaluate the oxidative damage of membrane. In conjunction with lipid degradation and peroxidation along with the fatty acid compositions from the initial stage of membrane lipid oxidation, the increased MDA content from the final stage of lipid oxidation comprehensively reflects the membrane damage in banana fruit related to chilling injury. Figure 6 suggests the possible mechanism of chilling injury of postharvest bananas related to lipid metabolism.

## 5. Conclusions

This study investigated the occurrence of chilling injury of banana fruit stored at low temperature with respect to membrane lipid metabolism. By application of lipidomic analysis, we systematically analyzed membrane lipids, fatty acids, lipid-related enzymes, and lipid peroxidation product in peel of bananas during storage at 6 °C for 0 and 4 d. Our results demonstrated that low temperature storage caused membrane lipid degradation and peroxidation, which consequently accounted for chilling injury symptoms in banana fruit. In this study, PC, PE, PA, DGDG, and MGDG were screened to be the biomarkers in chilling injury process. We also confirmed the lipid-related enzymes, PLD, and LOX were involved with the membrane degradation process in relation with production of ROS. In addition, the significant accumulation of PA, elevated ratio of DGDG to MGDG, and increased MDA content were observed in banana fruit with chilling injury. The probable mechanism of chilling injury of banana fruit-based membrane lipid metabolism was proposed.

## Figures and Tables

**Figure 1 foods-09-00894-f001:**
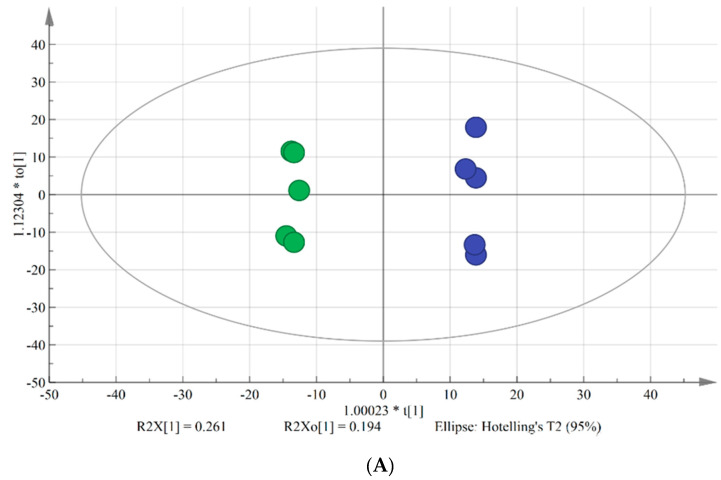
Score plot (**A**) and loading scatter plot (**B**) of OPLS-DA model. Green dots represent banana fruit stored for 0 d, and the blue dots represent banana fruit stored for 4 days at 6 °C.

**Figure 2 foods-09-00894-f002:**
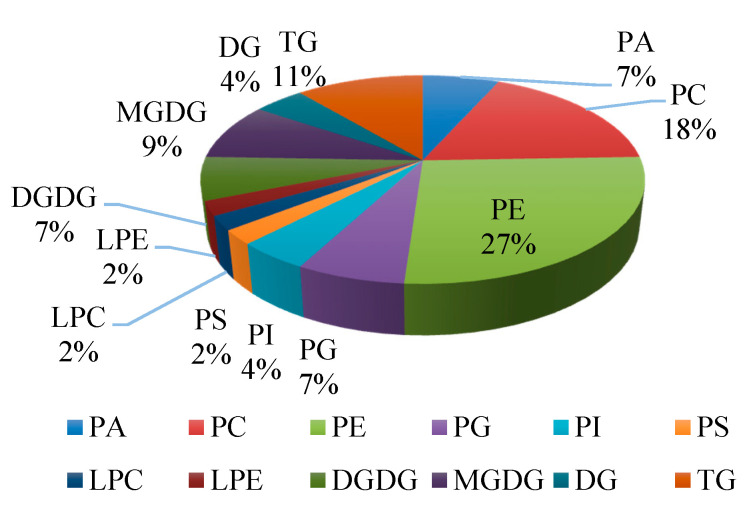
Classification of lipid species with significant differences of banana fruit during chilling injury. The proportions are based on the 45 lipids with significant differences. PA: Phosphatidic Acid; PC: Phosphatidylcholine; PE: Phosphatidylethanolamine; PG: Phosphatidylglycerols; PI: Phosphatidylinositol; PS: Phosphatidylserine; LPC: Lysophosphatidylcholine; LPE: Lysophosphatidylethanolamine; DGDG: Digalactosyldiacylglycerol; MGDG: Monogalactosyldiacylglycerol; DG: Diglyceride; TG: Triglyceride.

**Figure 3 foods-09-00894-f003:**
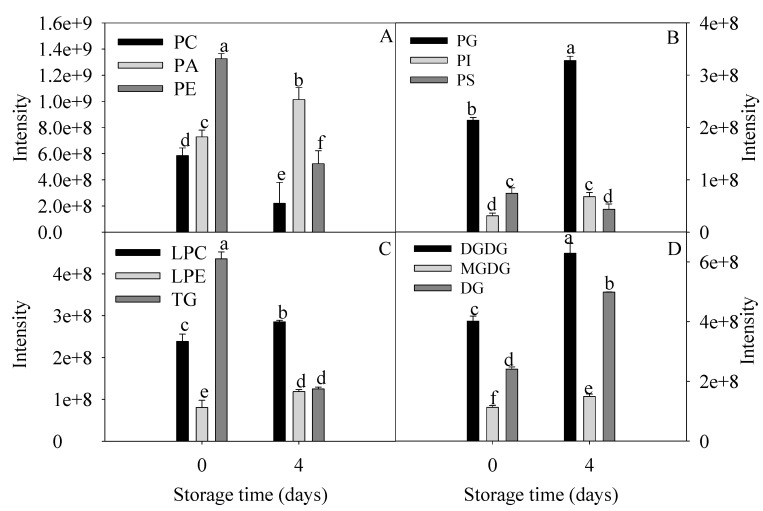
The relative content of lipid class with significant differences of banana fruit stored at 0 and 4 days at 6 °C. The content of each lipid species was presented as mean intensity ± standard error (*n* = 5). The means with different letters were significantly different (*p* < 0.05).

**Figure 4 foods-09-00894-f004:**
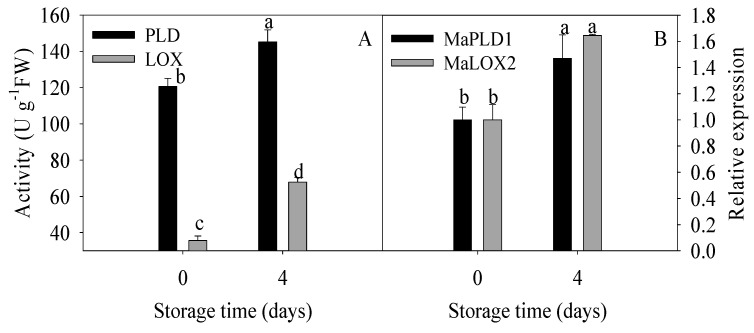
Enzymatic activities (**A**) and relative gene expressions (**B**) of phospholipase D (PLD) and lipoxygenase (LOX) in banana fruit stored for 0 and 4 d at 6 °C. The data were presented as means ± standard error (n = 3). The means with different letters were significantly different (*p* < 0.05).

**Figure 5 foods-09-00894-f005:**
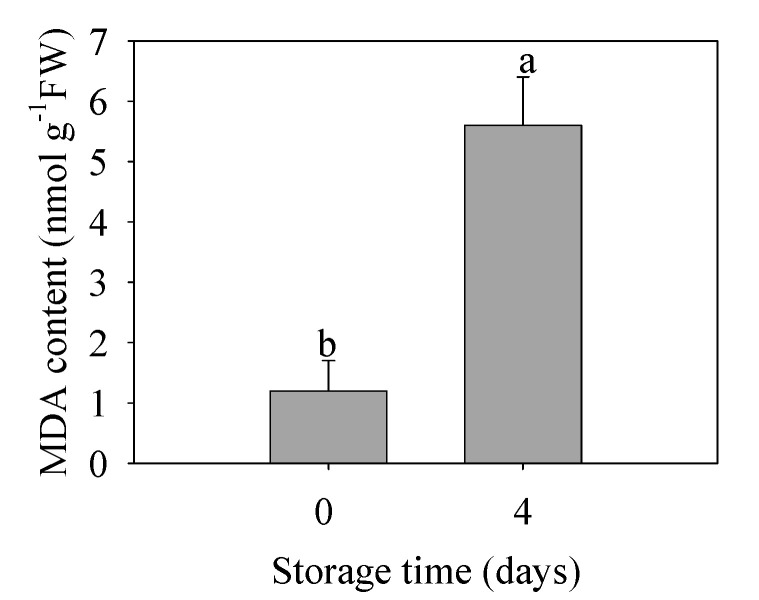
Changes in malondialdehyde (MDA) content in peel of banana fruit stored for 0 and 4 d at 6 °C. The data were presented as means ± standard error (*n* = 3). The means with different letters were significantly different (*p* < 0.05).

**Figure 6 foods-09-00894-f006:**
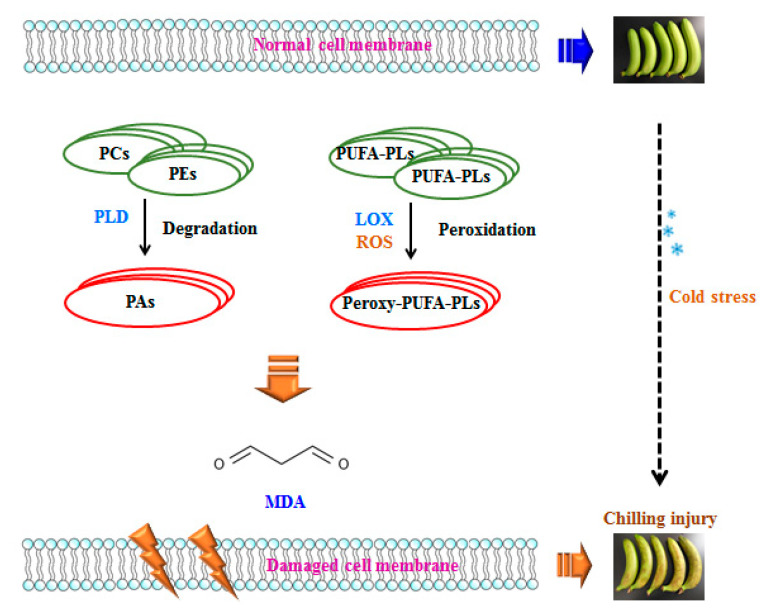
Probable mechanism of chilling injury of banana fruit based on membrane lipid metabolism. Phosphate group and hydrophobic chains of lipids are shown in sphere and line, respectively. PUFA: Polyunsaturated Fatty Acid; ROS: Reactive Oxygen Species.

**Table 1 foods-09-00894-t001:** The lipids with significant differences of banana fruit during chilling injury. VIP > 1 and *p* value < 0.05 were applied to determine these significantly different lipids.

Lipid Class	Fatty Acid	VIP	*p* Value	Fold Change
DG	(18:2/18:2)	1.40549	0.000164291	1.992905
DG	(18:3/18:2)	1.77349	0.000211777	2.122722
DGDG	(16:0/18:2)	1.53482	0.006909252	1.171822
DGDG	(16:0/18:3)	1.47239	0.019451923	1.217104
DGDG	(18:3/18:3)	1.17954	0.045065011	0.540866
MGDG	(18:1/18:2)	1.75277	0.021060252	0.399512
MGDG	(16:0/18:3)	1.73281	0.002636669	2.314542
MGDG	(18:2/18:2)	1.50627	0.040431337	5.129975
MGDG	(16:1/18:2)	1.07495	0.000376489	0.560368
TG	(18:3/18:2/18:3)	3.77031	0.023490239	0.074593
TG	(18:3/18:3/18:3)	3.59434	1.34298 × 10^−5^	0.320693
TG	(56:7)	3.00187	2.54755 × 10^−^^6^	0.207012
TG	(16:0/16:1/18:1)	1.64902	2.62766 × 10^−^^6^	7.914506
TG	(20:2/18:2/18:2)	1.22013	0.000935517	0.138964
**PA**	**(35:2)**	**4.22908**	**9.20627 × 10^−^** **^6^**	**0.360367**
PA	(18:2/18:2)	2.13718	0.01414934	1.637129
PA	(18:3/18:2)	1.02656	0.000960813	3.576563
**PC**	**(18:3/18:2)**	**11.9358**	**0.014104425**	**131.810267**
**PC**	**(18:3/18:3)**	**5.75108**	**0.0001374**	**1.872573**
PC	(14:0/18:2)	2.88833	0.022713176	0.761188
PC	(30:1)	2.24897	0.000328603	0.281843
PC	(32:4)	1.93317	0.022367818	0.529578
PC	(16:1/18:2)	1.71136	0.00109881	0.282682
PC	(16:0/16:1)	1.45801	8.65655 × 10^−5^	0.624131
PC	(38:4)	1.44706	0.004924833	55.163991
**PE**	**(16:0/18:2)**	**5.66391**	**0.015672899**	**2.638906**
PE	(16:0/18:3)	2.88624	0.038558	1.779774
PE	(18:3/18:3)	2.3183	3.19625 × 10^−5^	2.353246
PE	(24:0/18:2)	2.28216	0.014856478	4.593518
PE	(22:0/18:2)	1.9654	4.62275 × 10^−^^8^	25.075648
PE	(18:2/18:2)	1.95415	6.73907 × 10^−^^6^	0.283874
PE	(17:0/18:2)	1.39272	0.027108366	1.125154
PE	(18:0/18:2)	1.3809	0.042062853	1.182121
PE	(18:0/18:3)	1.31054	0.038018123	1.243902
PE	(18:0/18:3)	1.27068	0.003558509	1.423061
PE	(18:2/18:2)	1.18726	0.008143585	1.155013
PE	(24:0/18:3)	1.17305	0.012577722	6.053524
PI	(18:3/18:2)	1.45654	1.75695 × 10^−5^	2.424631
PI	(18:2/18:2)	1.18232	0.000420322	1.954515
PG	(16:0/18:2)	3.0636	0.002987396	2.016207
PG	(16:0/18:3)	1.40822	0.000845618	1.433207
PG	(40:6)	1.09057	0.001160027	1.211962
PS	(35:0)	1.46584	5.22552 × 10^−5^	0.585308
LPC	(18:2)	1.93911	0.017220137	0.483749
LPE	(18:2)	1.03778	0.002930842	0.479456

**Table 2 foods-09-00894-t002:** Ingredients and relative contents of fatty acids of membrane lipids in peel of bananas stored for 0 and 4 d at 6 °C. The data are presented as means ± standard errors (*n* = 3). The means with different letters are significantly different (*p* < 0.05).

Category	Fatty Acid Ingredient	Control (%)	Chilling Injury (%)
C_18:1_	Oleic acid	2.40 ± 0.08 ^a^	1.38 ± 0.10 ^b^
C_18:2_	Linoleic acid	4.35 ± 0.02 ^a^	3.27 ± 0.01 ^b^
C_18:3_	Linolenic acid	2.02 ± 0.03 ^a^	1.02 ± 0.12 ^b^
C_16:0_	Palmitic acid	32.11 ± 0.11 ^b^	36.65 ± 0.18 ^a^
C_18:0_	Stearic acid	25.34 ± 0.08 ^b^	29.88 ± 0.12 ^a^

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
