# Peer review of "Revealing Further Insights on Chilling Injury of Postharvest Bananas by Untargeted Lipidomics"

_foods, 2020, doi:10.3390/foods9070894_

Round 1

Reviewer 1 Report

This manuscript investigates the role of membrane lipid changes in chilling injury in bananas. The authors show that there are key changes in lipid composition due to chilling injury, with lipids such as PA and DGDG increasing, while PC and PE decrease. The authors also propose a mechanism for this change based on increased lipid peroxidation and degradation. While I believe that this paper provides an interesting and valuable addition to the field, there are several changes that I believe could be made to improve the quality of the paper. Throughout the paper there were some moderate errors in language and grammar that could be corrected. I also have the following specific concerns:

  1. To improve readability, please be sure to define all abbreviations on first use. For example, in line 66, LPC and LPE are used without previous definition. Line 64 and 75 also include abbreviations that weren’t used previously in the introduction (PA and PLD).
  2. Line 69: it is unclear to me what the authors are trying to say here. Are they indicating that PC, PE and PG are the main components of the membrane in peaches? Or are they the lipids that are primarily impacted by chilling injury in peaches?
  3. The methods include a statistical analysis section, but no description of the OPLS-DA model is included or how VIP values were obtained. It would be helpful if the authors could provide some discussion of how the model was applied in their study.
  4. Fig 1 appears to have low resolution, and the text is difficult to read. Also the y-axis in subplot C is not labeled.
  5. In Table 1, it would be helpful if the authors could indicate whether the changes are increases or decreases, perhaps using a positive or negative sign for the fold change. The organization of the table is also poor, making it very hard to find specific lipids within the table (such as the four lipids with large VIP values). It would be helpful to sort the table by lipid class, and the authors could also consider highlighting the largest changes using bold or italic text.
  6. It would be helpful if the authors could briefly discuss the difference between the VIP value and the p-value to highlight why lipids with high VIP values are more important than lipids with low p-values.
  7. In Fig 2 and the discussion for Table 1, the changes in lipids are discussed as percentages. It appears that these percentages are based on the number of lipid species that showed significant changes, but this should be clearly specified. While it is interesting to know the relative numbers of species changing, I think it may be even more informative to look at which lipid classes show the largest change overall. i.e., how does the overall lipid composition change, not just the number of lipid species change? Does the % composition of each lipid class change after chilling injury? All percentages should indicate whether they are based on number of lipid species or overall lipid composition. This confusion also occurs in line 249, where it refers to percentages of PC and PE in the control group.
  8. In Fig 4 the graphs not aligned, and the letters are a bit small on the graph, making it hard to see which values are significant.

Reviewer 2 Report

In the manuscript titled “Revealing Further Insights of Chilling Injury of Postharvest Bananas by Untargeted Lipidomics” Liu et al. describe their work on the study of changes in lipid content in banana exposed to low temperatures. Findings include decreases in PC and PE content, increased PA levels, as well as altered DGDG/MGDG ratios and changes in fatty acid profiles. These results are in line with previous studies in other species and present novel work as such studies have not been done with such detail in banana before. The experimentation appears sound and the manuscript is clearly written.

Although there are some grammatical mistakes, these do not detract from the message and should be easily corrected.

Please include a statement as to why the peels were analyzed and not the fruit flesh.

Please include a brief description added in the methods for the PLD and LOX activity assays.

Please include in the discussion the caveat that the lipid catabolic activity could also be the consequence rather than the cause of membrane damage. PLD activity has been shown to be greatly increased after the loss of membrane integrity (Bargmann et al. 2009; reference already used in the introduction), therefore the decrease of PC and PE and the increase in PA could also have occurred after cell lysis.
